# A Comparison of Raster-Based Forestland Data in Cropland Data Layer and the National Land Cover Database

Chinazor S. Azubike [1,*], Lyubov A. Kurkalova [2] and Timothy J. Mulrooney [3]

1   Applied Science & Technology Program, North Carolina A&T State University, Greensboro, NC 27411, USA
2   Department of Economics, North Carolina A&T State University, Greensboro, NC 27411, USA; lakurkal@ncat.edu
3   Department of Environmental, Earth and Geospatial Sciences, North Carolina Central University, Durham, NC 27707, USA; tmulroon@nccu.edu
*   Correspondence: csazubik@aggies.ncat.edu; Tel.: +1-615-710-6931

**Abstract:** The National Agricultural Statistics Service, the statistical arm of the US Department of Agriculture, and the Multi-Resolution Land Characteristics Consortium, a group of the US federal agencies, collect and publish several land-use and land-cover data sets. The aim of this study is to analyze the consistency of forestland estimates based on two widely used, publicly available products: the National Land-Cover Database (NLCD) and Cropland Data Layer (CDL). Both remote-sensing-based products provide raster-formatted land-cover categorization at a spatial resolution of 30 m. Although the processing of the yearly published CDL non-agricultural land-cover data is based on less frequently updated NLCD, the consistency of large-area forestland mapping between these two datasets has not been assessed. To assess the similarities and the differences between CDL- and NLCD-based forestland mappings for the state of North Carolina, we overlay the two data products for the years 2011 and 2016 in ArcMap 10.5.1 and analyze the location and attributes of the matched and mismatched forestland. We find that the mismatch is relatively smaller for the areas of the state where forests occupy larger shares of the total land, and that the relative mismatch is smaller in 2011 when compared to 2016. We also find that a large portion of the forestland mismatch is attributable to the dynamics of re-growth of periodically harvested and otherwise disturbed forests. Our results underscore the need for a holistic approach to data preparation, data attribution, and data accuracy when performing high-scale map-based analyses using each of these products.

**Keywords:** Geographic Information Systems (GISs); mapping; raster data; forestland; remote sensing; national land-cover database; cropland data layer; forest dynamics





## 1. Introduction

Large-area mapping of forest inventories is imperative for understanding forest ecosystems [1–6] and the economics of forest-based industries [7–11]. The majority of the analyses of North Carolina forestland use the data derived from the US Department of Agriculture Forest Service Forest Inventory and Analysis (FIA) surveys [7,12–16]. The program uses a stratified sample and defines forests from a use perspective, which, in general, refers to the human activities that alter land surface processes [17]. The use-based definition of forests relies on the interpretation of the conditions on the ground, such as a presence of a number of trees, at a point in time, with respect to intended use over a broader time [2]. Here, the intended use might be harvesting the trees for timber, or making sure that the trees remain undisturbed to ensure that the environmental amenities provided by the forest are intact.

An alternative definition of forest is from a land-cover perspective, which refers to the physical and biological cover over the surface of the land, such as water, vegetation, and bare soil [18]. The two major US programs providing periodic national assessments of land cover are the US Department of Agriculture (USDA) Cropland Data Layer (CDL) [19],

which are available yearly from 2008 for all US states, and National Land Cover Database (NLCD), which is produced by the Multi-Resolution Land Characteristics Consortium, a group of the US federal agencies, and is updated in approximately two-to-three-year intervals since 2001 [20–22]. In contrast with the use perspective, forest land-cover refers to the physical and biological cover over the surface of the land and, in addition to both grazed and ungrazed forests capable of being used for timber harvests, includes the forestland in parks, wildlife areas, and other special uses, where commercial timber harvests are rare [21,22].

In addition to defining forests from land-cover vs. land-use perspectives, CDL and NLCD differ from FIA, in that they both provide an opportunity to inventory forestland extent and location by assessing the entire population vs. only a sample of forests.

Recognizing the forest population representation, the NLCD is used as a basis for FIA stratification [17], and has been combined with auxiliary data providing, for example, tree height information, for the estimation of forest extent and stock [5,23–26]. However, actively managed forests undergoing periodic harvesting and replanting have been found to be interpreted as some other forms of land cover, such as shrub land and pasture/hay in the NLCD [6,26,27]. For that reason, more frequently produced data, such as the yearly CDL, may be helpful in improving the NLCD-derived understanding of forestland location and extent of harvesting and re-growth. However, only few studies juxtaposed the NLCD and CDL.

The use of CDL to specifically map forestland has been limited, and mostly focused on over-time conversions between forest and other land covers [28–32]. Early comparisons between CDL and NLCD data were focused on the cultivated cropland and were limited by mismatched timing of the collections [33–35]. The comparison of the 2008–2011 CDL with the 2006 NLCD revealed that, when the entire US is considered, 5.6% of the maps are in disagreement as to whether land is annually tilled for crops of not, with the latter category including forests [34]. In application to forests, ref. [36] found a 26% difference between 2005 CDL and 1992 NLCD forestland for Wisconsin, and [37] reported 8% and 1% differences between the 2001 NLCD and CDL (year(s) not specified) for North Dakota and South Dakota, respectively. A recent comparison of the maps derived from the 2011 NLCD and 2011 CDL for a 2.3 km$^2$ area in North Carolina revealed a 13% forest area difference between the two data sources [38].

The purpose of this article is to assess the usefulness of combining the NLCD and CDL for forestland mapping in an application to North Carolina. Specifically, we juxtapose the two databases for partially overlapping years to: (1) document the extent of mismatch between the same year CDL and NLCD in forestland mapping; and (2) evaluate the portion of the mismatch that can be attributed to the dynamics of re-growth of periodically harvested and/or otherwise disturbed forests. In the remainder of this article, we first introduce the study area and the NLCD and CDL data. Then, we describe our methods. Third, we present the results of the forestland assessment and mapping based on the NLCD and CDL for different regions of the state. Forth, we discuss the implications of our findings and conclude with potential future extensions of research.

## 2. Materials and Methods

The comparison of the CDL and NLCD datasets is a natural choice because both offer the complete coverage of the state, and share the same map projection, cell size, and reference grid origin.

### 2.1. Data

NLCD is an open-source land-cover database that provides spatially explicit national land-cover description at a spatial resolution of 30 m. Prior to 2019, four NLCD products were released in 1992, 2001, 2006, and 2011 [22]. The 2016 NLCD product, which was released in 2019, contained additional products, so that the revised NLCD collection provided land-cover maps at two-to-three-year intervals thorough the years 2001–2016 [21,39,40].

The data are a product of the Multi-Resolution Land Characteristics Consortium (MRLC), originally formed in 1993 to meet the needs of several federal agencies (US Geological Survey (USGS), Environmental Protection Agency, National Oceanic and Atmospheric Administration, and US Forest Service) [41], and are freely available through the MRLC website (https://www.mrlc.gov/, accessed on 9 June 2020). The details on the production and verification of NLCD data products are provided in [21,22,27,41].

North Carolina NLCD data are currently represented by a 16-class land-cover classification scheme rooted in [42] classification system. Of specific interest for this study are the four forestland cover categories (deciduous, evergreen, and mixed forests, and woody wetlands), and the categories that could represent or misrepresent re-growing forests, such as shrubs, grassland/herbaceous, pasture/hay, and cultivated crops (Table 1).

CDL is a 30 m resolution, raster-formatted, geo-referenced, crop-specific, land-cover map that utilizes ortho-rectified imagery to geospatially identify field crop types developed by the USDA NASS [19]. Field-level-resolution land cover data covering the conterminous 48 states is collected annually and publicly available [22]. The production and verification of the CDL data are detailed in [19,23].

CDL is a valuable tool for detecting change in land used for agriculture [22]. The CDL program covers the land used for certain crops, such as corn, soybeans, cotton, and wheat, and non-agricultural land with vegetation [19]. In comparison to other sources of land-use data, the spatially explicit identification of land use and land cover makes it a commonly used source for understanding agricultural land at a fine scale [23], as well as for detecting and analyzing the changes to and from agricultural and forestland uses [29,43].

North Carolina CDL has been produced every year (2008–2020). The data are available through an interactive data visualization portal CropScape, which provides open accessibility, visualization, and geospatial analytics to the user community (https://nassgeodata.gmu.edu/CropScape/, accessed on 5 September 2021). CDL is obtained through a supervised land-cover classification approach, which combines satellite imagery from sensors such as Landsat, Resourcesat, and the Disaster Monitoring Constellation [44].

CDL categories for water, developed land, barren, dryland forest, shrubland, and wetlands are in one-to-one agreement with the corresponding NLCD categories (Table 1). In general, the NLCD product is the source for non-cropland CDL processing [19], specifically ref. [22] point: "CDL does not simply revert to nor default to the NLCD in areas of non-cropland, but rather incorporates NLCD information for non-cropland as training data into the CDL's own, unique classification decision tree". In contrast with NLCD, CDL has a more detailed classification scheme for the herbaceous and planted/cultivated groups, bringing the total number of CDL categories to 116.

For the purposes of this analysis, we defined two new grouped land-cover categories, Forest and Grassland, which are, respectively, shown in darker and lighter shades of green in Table 1. Forest is the union of the four categories (deciduous, evergreen, and mixed forests, and woody wetlands). The Grassland definition differs between the data sets: it is the union of two NLCD categories (grassland/herbaceous and pastureland/hay), which is also the union of four CDL categories (alfalfa, other hay, sod/grass seed, and grassland/pasture).

The accuracy of NLCD forestland cover categories for the Eastern US has historically been high, with the 2011 user's accuracies for deciduous, evergreen, and mixed forests at a 92%, 85%, and 60% area, respectively. The user's accuracy was a 74% and 56% area for the woody wetlands and emergent herbaceous wetlands, respectively. The producer's accuracy for the same region and data release ranged between a 62% and 82% area for the dryland forest categories, and achieved an 87% and 76% area for the two wetland categories, respectively [45]. In general, the 2016 NLCD was evaluated to be as accurate as the 2011 NLCD [46]. Based on the recent assessments, the accuracy performance of CDL at the national scale for the categories of interest was similar to that of NLCD, and has improved during 2008–2016 [47].

**Table 1.** Selected land-cover categories in CDL and NLCD.

| Group * | NLCD Code and Category ** | NLCD Definition *** | CDL Code and Category **** |
|---|---|---|---|
| Dryland forest | 41 Deciduous Forest | Areas dominated by trees generally greater than 5 m tall, and greater than 20% of total vegetation cover. More than 75% of the tree species shed foliage simultaneously in response to seasonal change. | 141 Deciduous Forest |
| | 42 Evergreen Forest | Areas dominated by trees generally greater than 5 m tall, and greater than 20% of total vegetation cover. More than 75% of the tree species maintain their leaves all year. Canopy is never without green foliage. | 142 Evergreen Forest |
| | 43 Mixed Forest | Areas dominated by trees generally greater than 5 m tall, and greater than 20% of total vegetation cover. Neither deciduous nor evergreen species are greater than 75% of total tree cover. | 143 Mixed Forest |
| Shrubland | 52 Shrubland | Areas dominated by shrubs less than 5 m tall with shrub canopy typically greater than 20% of total vegetation. This class includes true shrubs, young trees in an early successional stage, or trees stunted from environmental conditions. | 152 Shrubland |
| Herbaceous | 71 Grassland/ Herbaceous | Areas dominated by graminoid or herbaceous vegetation, generally greater than 80% of total vegetation. These areas are not subject to intensive management, such as tilling, but can be utilized for grazing. | 36 Alfalfa; 37 Other Hay; 59 Sod/Grass Seed; 176 Grassland/ Pasture |
| Planted/ Cultivated | 81 Pasture/Hay | Areas of grasses, legumes, or grass/legume mixtures planted for livestock grazing or the production of seed or hay crops, typically on a perennial cycle. Pasture/hay vegetation accounts for greater than 20% of total vegetation. | |
| | 82 Cultivated Crops | Areas used for the production of annual crops, such as corn, soybeans, vegetables, tobacco, and cotton, and also perennial woody crops, such as orchards and vineyards. Crop vegetation accounts for greater than 20% of total vegetation. This class also includes all land being actively tilled. | Multiple cultivated crop categories |
| Wetlands | 90 Woody Wetlands | Areas where forest or shrubland vegetation accounts for greater than 20% of vegetative cover, and the soil or substrate is periodically saturated or covered with water. | 190 Woody Wetlands |
| | 95 Emergent Herbaceous Wetlands | Areas where perennial herbaceous vegetation accounts for greater than 80% of vegetative cover, and the soil or substrate is periodically saturated or covered with water. | 195 Herbaceous Wetlands |

Notes: * Classes not shown: water, developed land, and barren. ** Categories that do not apply to North Carolina, e.g., perennial ice/snow, are not shown. *** Source: https://www.mrlc.gov/data/legends/national-land-cover-database-2016-nlcd2016-legend, accessed on 11 May 2021. **** Source: https://nassgeodata.gmu.edu/CropScape/, accessed on 11 May 2021. In this study, the darker shaded categories are referred to collectively as Forest, and the lighter shaded categories are referred to collectively as Grassland.

### 2.2. Study Area

The current study addressed the state of North Carolina. North Carolina was chosen for this study because of the complexity of the state's land use/land cover. North Carolina is one of the most physio-geographically diverse states in the southern United States comprising of the Coastal Plain in the eastern part of the state, the Piedmont in the central

part of the state, and the mountains in the west [15]. For statistical reporting purposes, USDA NASS divides North Carolina into eight Agricultural Statistical Districts (ASDs), as shown in Figure 1. ASDs are groups of counties, distinguished by climate (temperature and annual precipitation), geography (soil type, terrain, and elevation), and crop production practices [37]. According to the latest available North Carolina Forest Inventory Analysis (FIA) inventory cycle completed in 2013, forests occupied approximately 60% of the state's land between 2007 and 2013, approximately 18.6 million acres. Both CDL and the NLCD estimate smaller forestland areas, when compared to the 2007-13 FIA: 16.8 million acres or 55% of the state total according to 2011 NLCD and 18.1 million acres or 59% of the state total according to the 2011 CDL. FIA surveys divide the state into four regions (units), with the percent of forested area varying between 51% in the Central Piedmont unit, and 76% in the Western Mountains unit. The CDL and NLCD reveal a similar general geographical pattern of forestland, with the highest proportion of forestland in total appearing in the two Mountain ASDs (Figure 1).

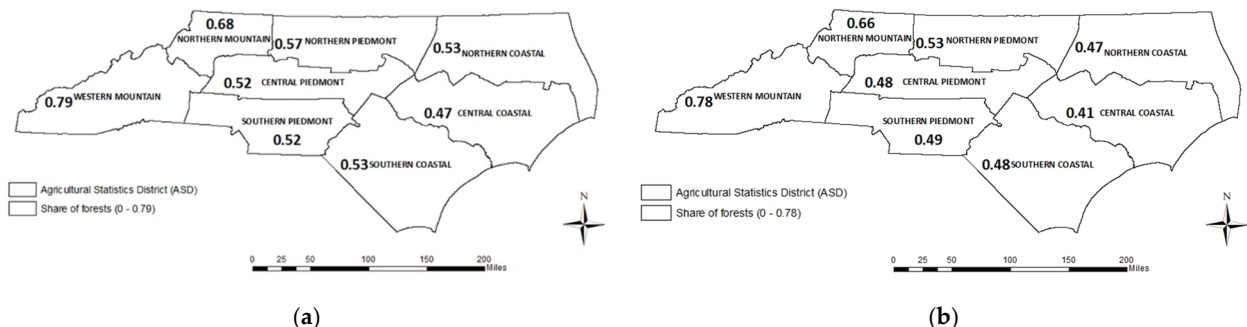

**Figure 1.** Map of Agricultural Statistical Districts (ASDs) in North Carolina based on the share of forestland in CDL (**a**) and share of forestland in the NLCD (**b**).

*2.3. Methods*

We juxtaposed the 2011 NLCD and CDL and the 2016 NLCD and CDL in ArcMap 10.5.1 to identify the matched and mismatched forestland pixels for each ASD in the state. GIS data in raster format were downloaded from CDL (https://nassgeodata.gmu.edu/CropScape/, accessed on 12 November 2019) and NLCD (https://www.mrlc.gov/data, accessed on 12 November 2019) data repositories. The processes involved are summarized in Figure 2. The data were projected to the NC State Plane (US Feet) projection. The North Carolina data layer was extracted by mask using an NC county layer map available through NC One Map (http://www.nconemap.gov, accessed on 12 November 2019). The CDL and NLCD layers were reclassified to a general forest category. The forest classes in CDL (deciduous forest = 41, evergreen forest = 42, mixed forest = 43, and woody wetlands = 90) and the forest classes in NLCD (deciduous forest = 141, evergreen forest = 142, mixed forest = 143, and woody wetlands = 190) were classified as 1, while other classes (for example, agricultural land, water, and developed land) were classified as 0, and NoData (where no data existed) was left as NoData. The new forest layer was overlaid on an NC OneMap Ortho-imagery layer to provide aesthetics. This was conducted to illustrate the spatial distribution of forest land-cover in the study area.

Map Algebra was used to calculate the quantitative inconsistencies between the datasets. In this case, the newly created general forest data layer (1 = forest, 0 = non-forest) for CDL was subtracted from the general forest layer for the NLCD, and the result of this Map Algebra operation could be −1, 0, or 1. Here, −1 represents forest in the NLCD but a different category in CDL, 0 represents where they both match, and 1 represents forest in CDL but a different category in the NLCD. All values of −1 (forestland in the NLCD) and 1 (forestland in CDL) were extracted to their own layer using the *Reclassify* function. These new layers, whose values contained 1 or NoData, were multiplied by the original CDL or NLCD layers to render a layer of the summary of their pixels and locations of inconsistencies.

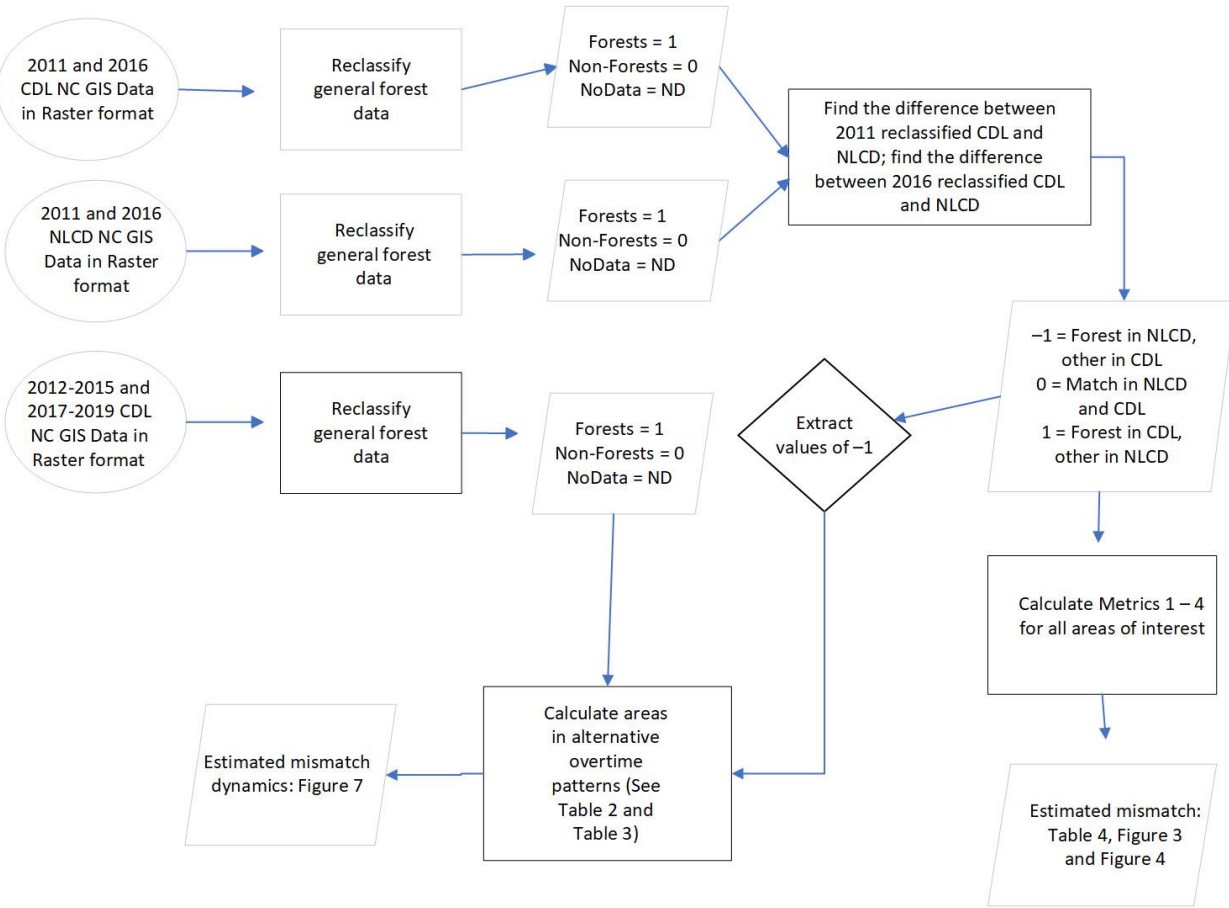

**Figure 2.** Flow diagram of data development and analysis.

The two snapshots of the differences in 2011 and 2016 were each quantified in a total of four metrics, separately for each ASD. Two metrics reflected the absolute forestland mismatch: Metric 1 was the total area identified as forestland in CDL and something else in the NLCD, i.e., coded 1, and Metric 2 was the total area identified as forestland in the NLCD and something else in CDL, i.e., coded $-1$. Since the total and forestland areas varied significantly across the state, to facilitate between-region comparisons, we also considered two relative measures of mismatch. The mismatch relative to match was calculated as the absolute forestland mismatch divided by the area identified as forest in both data sets, for the two metrics of the absolute mismatch separately. Specifically, we calculated Metric 3 as the percentage of the total area identified as forestland in CDL and something else in NLCD, i.e., coded 1, in the total area coded 0, i.e., identified as forestland in both datasets. We calculated Metric 4 as the percentage of the total area identified as forestland in the NLCD and something else in CDL, i.e., coded $-1$, in the total area coded 0, i.e., identified as forestland in both datasets.

To analyze the mismatch over time, we traced how the land that was mismatched was recorded in CDL in the years subsequent to 2011 (2012–2015) and in the years subsequent to 2016 (2017–2019). For the years for which only CDL was available, i.e., for the years 2012–2015 and 2017–2019, we first *Reclassified* the forest classes (deciduous forest = 41, evergreen forest = 42, mixed forest = 43, and woody wetlands = 90) as 1, while other classes (such as agricultural land, water, and developed land) were reclassified as 0, and left the NoData (where no data existed) as NoData. Subsequently, only for the land identified as forestland in the NLCD and something else in CDL, i.e., coded $-1$ in 2011 and 2016, we used *Map Algebra* to multiply the reclassified CDL data by $-1$. We then analyzed what

portion of the total mismatch in 2011 and 2016 fell in the categories that identified forestland as described in the patterns presented in Tables 2 and 3, separately for each ASD.

**Table 2.** Tracing the land identified as forest in the 2011 NLCD, but identified as non-forest in the 2011 CDL using the subsequent year's CDL data.

| Reclassified CDL Coding: −1—Forest, 0—Non-Forest | | | | Identification of Forestland |
|---|---|---|---|---|
| **2012** | **2013** | **2014** | **2015** | |
| 0 | 0 | 0 | −1 | Land identified as non-forest in 2012, 2013, and 2014, but is identified as forest in 2015 |
| 0 | 0 | −1 | −1 | Land identified as non-forest in 2012 and 2013, but is identified as forest from 2014 |
| 0 | −1 | −1 | −1 | Land identified as non-forest in 2012, but is identified as forest from 2013 |
| −1 | −1 | −1 | −1 | Land identified as forest in 2012–2015 |

**Table 3.** Tracing the land identified as forest in 2016 NLCD, but identified as non-forest in 2016 CDL using the subsequent years CDL data.

| Reclassified CDL Coding: −1—Forest, 0—Non-Forest | | | Identification of Forestland |
|---|---|---|---|
| **2017** | **2018** | **2019** | |
| 0 | 0 | −1 | Land identified as non-forest in 2017 and 2018, but is identified as forest in 2019 |
| 0 | −1 | −1 | Land identified as non-forest in 2017, but is identified as forest from 2018 |
| −1 | −1 | −1 | Land identified as forest in subsequent years of CDL |

The re-coded CDL and NLCD data were used to answer two research questions and to test two hypotheses for each question.

Research question 1: how large is the NLCD–CDL mismatch in forest representation in NC for the same years 2011 and 2016?

**Hypotheses 1.1.** *Forestland mismatch is relatively minor and varies across the ASDs, with the mismatch negatively correlated with the share of forestland in total.*

**Hypotheses 1.2.** *Forestland mismatch is smaller in 2016 when compared to 2011.*

Research question 2: could the mismatch be attributed to the dynamics of re-growth of periodically harvested and/or otherwise disturbed forests?

**Hypotheses 2.1.** *A large portion of the mismatch is attributable to the dynamics of re-growth and re-planting, i.e., it is a part of the cycle of forest to grass to shrubland back to forest then grass.*

**Hypotheses 2.2.** *The mismatch is qualitatively of the same dynamics for 2011–2015 and 2016–2019.*

## 3. Results

*Research question 1*: how large is the NLCD–CDL mismatch in forest representation in NC for the same years 2011 and 2016? The absolute mismatch in forestland areas by ASD is detailed in Table 4, and the relative mismatch—in both Table 4 and Figure 3.

Both Table 4 and Figure 3 reveal support for *Hypothesis 1.1* that suggests that the forestland mismatch as a percent of match varies between 3% and 27% for the eight ASDs assessed in 2011 and 2016, and the relative mismatch is consistently smaller in the Mountain ADSs.

**Table 4.** Agricultural Statistics Districts, the total area for each (ha), total forestland that match between CDL and NLCD in 2011, and the percent relative to match.

| | Southern Coastal | Central Coastal | Northern Coastal | Southern Piedmont | Central Piedmont | Northern Piedmont | Western Mountain | Northern Mountain |
|---|---|---|---|---|---|---|---|---|
| Total Area (Ha) | 2,390,868 | 1,721,521 | 1,684,806 | 1,377,242 | 1,358,993 | 1,504,857 | 1,881,924 | 862,955 |
| **2011 Comparison** | | | | | | | | |
| Total Forest Match 2011 (Ha) | 1,039,990 | 635,858 | 733,571 | 1,133,985 | 628,472 | 764,397 | 1,433,947 | 553,434 |
| Metric 1: Total Forest in CDL, something else in NLCD (Ha) | 228,941 | 166,548 | 156,356 | 86,300 | 74,578 | 90,563 | 54,528 | 33,634 |
| Metric 3: Relative to Match (%) | 22% | 26% | 21% | 8% | 12% | 12% | 4% | 6% |
| Metric 2: Total Forest in NLCD, something else in CDL (Ha) | 102,099 | 69,104 | 61,181 | 53,389 | 24,597 | 33,935 | 37,199 | 14,989 |
| Metric 4: Relative to Match (%) | 10% | 11% | 8% | 5% | 4% | 4% | 3% | 3% |
| **2016 Comparison** | | | | | | | | |
| Total Forest Match 2016 (Ha) | 1,111,215 | 655,016 | 735,967 | 596,482 | 635,510 | 773,195 | 1,420,935 | 551,529 |
| Metric 1: Total Forest in CDL, something else in NLCD (Ha) | 135,043 | 101,413 | 97,527 | 141,547 | 88,822 | 107,900 | 100,895 | 58,252 |
| Metric 3: Relative to Match (%) | 12% | 15% | 13% | 24% | 14% | 14% | 7% | 11% |
| Metric 2: Total Forest in NLCD, something else in CDL (Ha) | 236,042 | 175,180 | 161,984 | 115,354 | 62,371 | 89,170 | 60,015 | 29,603 |
| Metric 4: Relative to Match (%) | 21% | 27% | 22% | 19% | 10% | 12% | 4% | 5% |

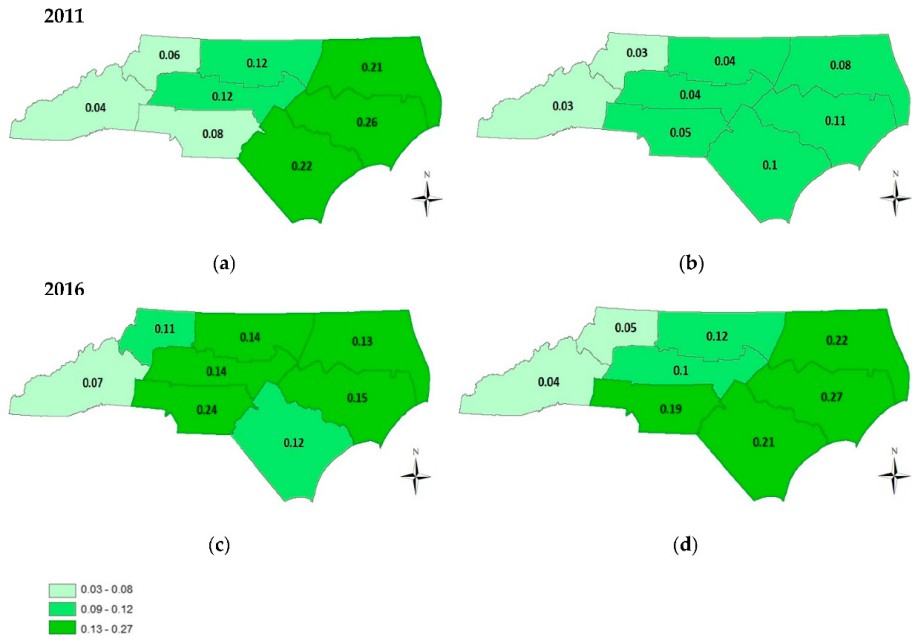

**Figure 3.** Mismatch as a percentage of match: (**a**) forestland in CDL, but something else in the NLCD in 2011; (**b**) forestland in the NLCD, but something else in CDL in 2011; (**c**) forestland in CDL, but something else in the NLCD in 2016; (**d**) forestland in NLCD, but something else in CDL in 2016.

Unlike for *Hypothesis 1.1*, we do not find consistent support for *Hypothesis 1.2*. As Table 4 and Figure 3 detail, both the absolute and the relative forestland mismatch are less in 2016 than in 2011 only in Metric 1 (forestland in CDL, other land in NLCD) and only for one region: the Coastal ASDs.

To formally test *Hypothesis 1.1*, we calculated the correlation coefficients between the share of forests and Metrics 3 and 4, using the ASDs as the units of analysis (eight ASDs, two years, a total of eight units of analysis) (Table 5), and formally tested the following hypotheses:

**Table 5.** Correlation table showing the relationship between the share of forests and forestland mismatch.

| Mismatch | Share of Forest in Total Area Based on 2011 NLCD | Mismatch | | | |
|---|---|---|---|---|---|
| | | *Metric 3: Forest CDL, Other NLCD 2011* | *Metric 4: Forest NLCD, Other CDL 2011* | *Metric 3: Forest CDL, Other NLCD 2016* | *Metric 4: Forest NLCD, Other CDL 2016* |
| *Metric 3:* Forest CDL, other NLCD 2011 | −0.79 | 1 | | | |
| *Metric 4:* Forest NLCD, other CDL 2011 | −0.71 | 0.95 | 1 | | |
| *Metric 3:* Forest CDL, other NLCD 2016 | −0.60 | 0.08 | 0.14 | 1 | |
| *Metric 4:* Forest NLCD, other CDL 2016 | −0.85 | 0.88 | 0.91 | 0.49 | 1 |

**H0.** $\rho = 0$

**Ha:** $\rho < 0$

Here, $\rho$ is the correlation between the share of forests and a metric of forestland mismatch. Under conventional assumptions, at a 5% level of significance, we could not reject the null hypothesis for one measure: Metric 3 (forest CDL, other NLCD 2016). However, we rejected the null hypothesis in favor of the alternative for the other three correlation coefficients considered, at a 5% level of significance; that is, we found support for *Hypothesis 1.1* about the mismatch being less for the ASDs with the higher share of forestland in total.

*Research question 2*: could the mismatch be attributed to the dynamics of re-growth of periodically harvested and/or otherwise disturbed forests? To test *Hypothesis 2.1*, we analyzed the composition of the absolute mismatch, with a specific focus on the land cover that could be attributed to the transitions within a forest cuts and re-growth cycle. It has been shown that the vegetation in place of cut trees could show as grasses or shrubland when the remote sensing techniques are used to gather data [6,45,48]. Thus, for the purposes of testing *Hypothesis 2.1*, we postulated that the forest mismatch could be attributed to the dynamics of re-growth if shrubland and grasses combined constitute more than half of the mismatch. As Figure 4 reveals, under such a rule, the hypothesis is supported in the Piedmont for both metrics and both years, but is not supported in the other regions of the state with at least one metric. The pie charts show the composition of the absolute mismatch, and shaded areas show the ASDs for which we do not find support for the *Hypothesis 2.1*.

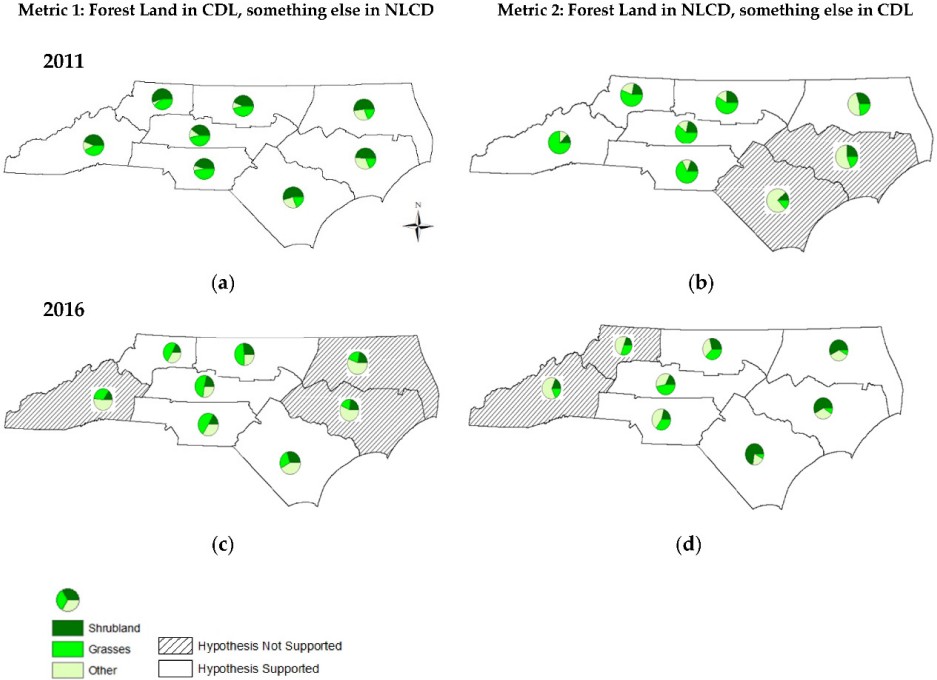

**Figure 4.** (**a–d**) Mismatched forestland in 2011 and 2016. Pie charts show the composition of the absolute mismatch, and shaded areas show the ASDs that do not support *Hypothesis 2.1*.

To test *Hypothesis 2.2*, we analyzed the over-time composition of the absolute mismatch in Metric 2, forestland in the NLCD, and something else in CDL. Specifically, we analyzed what share of such land was recorded as forest in subsequent years of CDL. As Figure 5a shows, for all the ASDs considered, over 30% of the 2011 mismatched land is identified as forest in 2012, with the 2013–2015 period showing a steady increase in the share of the mismatched land being identified as forest. The highest rate of 2013–2015 increase, 11%, is in the Central Piedmont (Figure 5a), and the percentage of the mismatch recorded as forest exceeds 50% by the year 2015 for all ASDs.

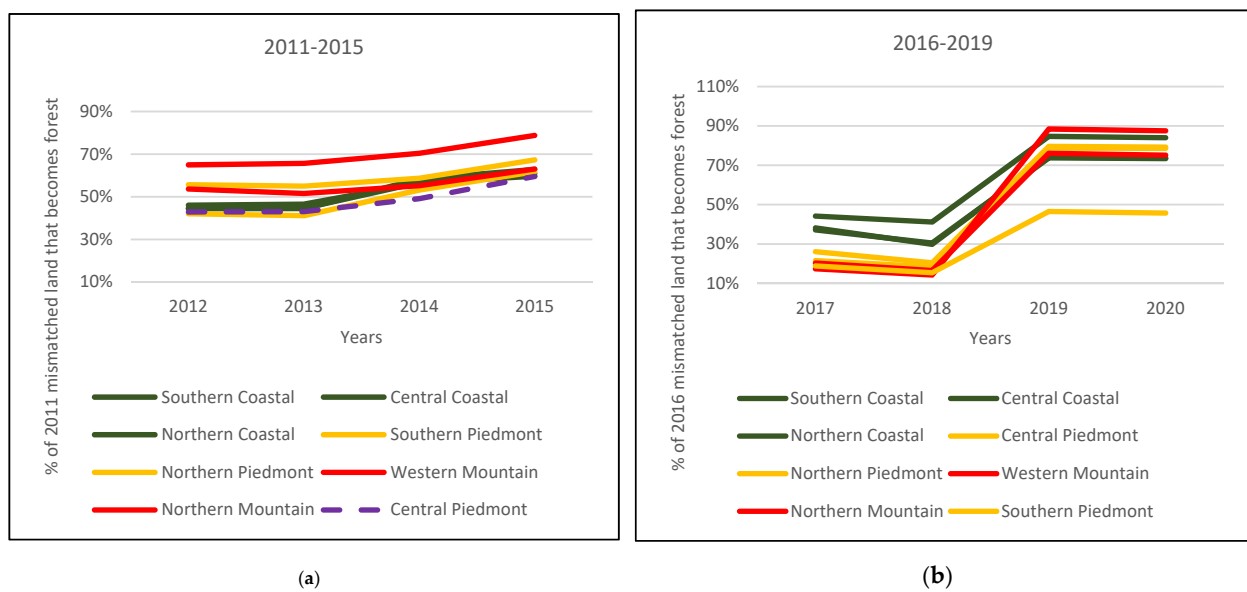

**Figure 5.** Land identified as forest in the NLCD, but identified as non-forest in in CDL that becomes forest in CDL in subsequent years of data availability; (**a**) percent of mismatched land in 2011 that becomes forest; (**b**) percent of mismatched land in 2016 that becomes forest.

The dynamics of the forest identification in the 2016 mismatch are qualitatively the same as those for the 2011 mismatch, in that a large portion of the mismatch is recorded as forest three years later. However, in Figure 5b, there is a "jump" in the percent of land that is identified as forest between 2018 and 2019, most notably a 76% increase in the Western Mountain ASD and a 47% increase in the Southern Coastal ASD.

## 4. Discussion

The NLCD and CDL are both rich sources for high-quality land-cover data. Data are provided at regular intervals, with CDL updates released yearly, and the NLCD—in five-year intervals since 2001. Both data sets experience rigorous accuracy checks [18,19]. This study is the first to compare and contrast the forestland maps derived from the two sources for the state of North Carolina, where forest occupy approximately 60% of the land's surface.

Overall, we found that the disagreement between the maps for which both sets of data were available, 2011 and 2016, was relatively small, and was the smallest in the Mountains where the share of forestland in total was the highest when compared to the Piedmont or Coastal regions. The only known previous direct comparison of the 2011 CDL and NLCD that was for a random 2.3 km$^2$ area in the Southern Coastal Agricultural Statistical District region of North Carolina, yielded estimates that were similar to ours: 12% and 10% in Metrics 3 and 4, respectively [38].

More generally, when the eight ASDs were considered, we found the relative mismatch decreasing with forest predominance on the landscape. For example, the coefficient of correlation between the ASD share of forest in total area (based on 2011 NLCD) and any of the metrics of mismatch considered in the study was −0.60 or lower, all statistically significant at a 5% level of significance. We posit that this finding is likely to hold for other geographic regions, though additional research may be needed to evaluate, for example, how high the share of forestland in total needs to be to confidently assume that the relative forestland mismatch between the two datasets is less than a desired threshold.

The global-scale model comparison conducted by Prestele and collaborators [49] showed that the differences in input data applied to simulation models constitute an importance source of uncertainty in forest areas. In such a context, our analysis contributes to the understanding of the uncertainty of the data that are commonly used as input to the evaluation of the land-cover change in the US and the associated environmental impacts [45]. Another area of research where such an evaluation of the consistency of the alternative input data are of importance, concerns the assessment of forest biomass and the production of renewable energy [38].

Our analysis also found that the support of the hypothesis that a large share of the mismatch between the two data sources was attributable to the dynamics of forest cuts or other disturbances and regrowth. Again, we found that the rate at which the mismatched land was identified as forest in subsequent CDL releases varied across space. An intriguing area of future research would be to identify other land-cover attributes, such as the degree of forest fragmentation, land-cover clustering, or proximity to population centers, which would help explain and predict the changes in the mismatch overtime. Supplementing the land-cover maps with the airborne or satellite high-resolution data for the identification of individual trees on the mismatched land could likewise be helpful for understanding the reasons behind the mismatch [50].

This paper quantified the level of disagreement between the widely used data products, CDL and NLCD. GIS techniques utilizing change detection analysis were used to quantitatively measure this level of agreement. This is not to say that the data are wrong, but only that assuming that NLCD forest matches with CDL forest with 100 percent accuracy is incorrect. Our findings point to the need to be apprehensive about potential land-cover identification errors when using these remote-sensing-based data for forestland assessments.

The results shed light on the opportunities and potential pitfalls associated with using NLCD and CDL for large-area forestland assessment and mapping. Unlike the sample-

based FIA, CDL and NLCD provide an opportunity to inventory forestland by assessing the entire population vs. only a sample of forests. The development of the analytical approaches that combine the sample-based and population-based data for forestland mapping is likely to improve future assessments of the environmental and economic assessments of forest inventories.

## 5. Conclusions

By utilizing rich data sets representing land cover with varying granularities and created from different sources, the use and application of GIS techniques can be used to perform change detection analysis to measure the level of agreement between NLCD and CDL. As highlighted by Figure 2, raster GIS techniques of reclassify-and-map algebra provide measurements of change detection to make comparisons between forest and non-forest in these datasets.

Forestland misclassification could impact policy-making as related to forests and related industries. For example, a better understanding of the presence of forests will help the regional offices that handle and allocate funds from certain programs, such as the North Carolina Forest Stewardship Program (NCFSP) (https://www.ncforestservice.gov/, accessed on 24 March 2022). The NCFSP is a voluntary cooperative effort where private landowners receive technical assistance in developing a management plan to help them achieve their objectives. They are also recognized for their achievements in promoting forest resource management and are connected with the information and tools they need to manage their forests and woodlands. Better understanding the spatial distribution of the forestland that is eligible for the NCFSP would be helpful for the program funding planning. Similarly, the comparisons of forestland in CDL and NLCD could benefit the planning of other federal forest-related policies, such as the Forest Land Enhancement Program (FLEP), Forest Legacy Program (FLP), and the Forestry Incentives Program (FIP) (https://www.fs.usda.gov/managing-land, accessed on 24 March 2022).

Our analysis of North Carolina forestland-based mapping contributes to the literature in two important ways. First, we documented the comparison of forestland mapping for NC. Second, we explored the utility of juxtaposing the multiple years of CDL and NLCD for enhancing the dynamic information about the forests. Our findings do not suggest that one product is more accurate or should be preferred for forest mapping. We found that the relative mismatch was small and varied across the state, and that a large portion of the forestland mismatch was attributable to the dynamics of re-growth of periodically harvested and otherwise disturbed forests. These findings imply that the best practices would be to report the estimates based on both CDL and NLCD forestland maps, when practical and possible. While this study focused on identifying and quantifying the differences in the two data products, future research could focus on developing the methods for merging the two maps for forest assessments.

Future forest area mapping and assessment could combine the CDL and NLCD with other developing land-cover and land-use monitoring products ranging from those started by non-government organizations, such as the Global Forest Watch (https://www.globalforestwatch.org, accessed on 24 March 2022), to rigorous scientific efforts by the government agencies. For example, the ongoing US Geological Survey Land Change Monitoring, Assessment, and Projection (LCMAP) project (https://www.usgs.gov/special-topics/lcmap, accessed on 24 March 2022) generates an integrated collection of annual land-cover products for the conterminous United States with the specific focus on documenting the change in land surface [25]. We did not include the LCMAP data in our analysis because they are not independent from NLCD, in that the 2011 NLCD are used as training data for LCMAP classification and for post-classification gap-filling (USGS, 2021). Similar to the LCMAP, the Landscape Change Monitoring System (LCMS) produced by the US Forest Service of USDA centers on modeling and mapping change, but with a focus that is closer to the topic of our analysis: the change in the vegetation cover. As with the LCMAP, various data products, including NLCD, are used as inputs to modeling [49]. Future research that

juxtaposes the NLCD–CDL forest mismatch maps with the LCMAP and LCMS output products, such as the maps that represent alternative stages of after-disturbance forest recovery, could provide additional insights into forest dynamics.

Finally, our study focused on two continuous US land-cover mapping programs. However, the accurate representation of forestland worldwide is likewise important for multiple reasons, such as greenhouse gas reduction efforts through reducing deforestation and reforestation [30,51–56]. Our findings point to the need to be mindful about the potential mismatch in the identification of forest land from alternative land-cover products, especially when identifying periodically harvested and otherwise disturbed forests.

**Author Contributions:** Conceptualization, C.S.A. and L.A.K.; methodology, C.S.A. and T.J.M.; formal analysis, C.S.A.; investigation, C.S.A. and T.J.M.; data curation, C.S.A.; writing—original draft preparation, C.S.A., L.A.K. and T.J.M. All authors have read and agreed to the published version of the manuscript.

**Funding:** This study was funded by Research Partnership Agreement 15-JV-11330143-010 between USDA Forest Service and NCA&T and the Title III HBGI grant funded by the U.S. Department of Education.

**Institutional Review Board Statement:** Not applicable.

**Informed Consent Statement:** Not applicable.

**Data Availability Statement:** Publicly available datasets were analyzed in this study. This data can be found here: www.mrlc.gov/data and https://nassgeodata.gmu.edu/CropScape/, accessed on 16 May 2022.

**Conflicts of Interest:** The authors declare no conflict of interest.

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
