# Peer review of "A Comparison of Raster-Based Forestland Data in Cropland Data Layer and the National Land Cover Database"

_forests, doi:10.3390/f13071023_

Round 1

Reviewer 1 Report

Summary: this paper documents the differences between the National Land Cover Dataset and the Cropland Data Layer in regards to forest cover mapping in North Carolina. Statistics are generated at the state and USDA statistic districts level.

Crop Data Layer should be Cropland Data Layer

NLCD remaps more frequently than every five years.

Does the CDL publish statistics on its accuracy/uncertainty of its forest classes? Do you think USDA would suggest the CDL should even be used for forest monitoring?

Why not also consider the products from LCMAP? It was highlighted in the conclusion after all.

I don’t think the technical detail about the making of the NLCD and CDL products are needed. Just reference them and move on.

The comparison of the NLCD and CDL against each other could be interesting to some readers. However, most people what to know what product is best to use in terms of accurately depicting what is on the ground. That was not provided. If a combined map is best then it would be nice to see a method to merge them and then show its accuracy (with some sort of in situ data).

Author Response

  1. Crop Data Layer should be Cropland Data Layer

Thank you. This has been corrected in text.

  1. NLCD remaps more frequently than every five years.

Thank you for pointing out to this relatively recent change in NLCD time step, which was implemented with the 2016 NLCD product released in 2019. We revised the text in the Abstract, Introduction, and Materials and Methods to clarify this.

  1. Does the CDL publish statistics on its accuracy/uncertainty of its forest classes? Do you think USDA would suggest the CDL should even be used for forest monitoring?

The USDA does not publish the statistics on the accuracy/uncertainty of the CDL forest classes. As we detail in the introduction, being more frequently (yearly) updated land cover data, the CDL is commonly used for the analyses of land use/land cover change. We revised the text in the Materials and Methods section to emphasize that the CDL is more commonly used to understand agricultural land at a fine scale, as well as for detecting and analyzing the changes to and from agricultural and forest land. Thank you.

  1. Why not also consider the products from LCMAP? It was highlighted in the conclusion after all.

As we detailed in the Conclusions, we did not include the LCMAP data in our analysis because they are not independent from NLCD in that the 2011 NLCD are used as training data for LCMAP classification and for post-classification gap-filling. We also point that future research that juxtaposes the NLCD-CDL forest mismatch maps with the LCMAP could provide additional insights about forest dynamics. Thank you.

  1. I don’t think the technical detail about the making of the NLCD and CDL products are needed. Just reference them and move on.

We followed your advice and removed the technical details on NLCD and CDL productions in the Materials and Methods section. Thank you.

  1. The comparison of the NLCD and CDL against each other could be interesting to some readers. However, most people what to know what product is best to use in terms of accurately depicting what is on the ground. That was not provided. If a combined map is best then it would be nice to see a method to merge them and then show its accuracy (with some sort of in situ data).

We agree with the comment. We added a comment in the Conclusions section pointing that while this study focused on identifying and quantifying the differences in the two data products, future research could focus on developing the methods for merging the two maps for forest assessments. Thank you.

Reviewer 2 Report

Overall paper good and I have accepted for publication with some minor changes and author should add following references in the line. no.138 in material methods section.

1. Chaitanya B. Pande, Kanak N. Moharir, Sudhir Kumar Singh, Abhay M.Varade ,Ahmed Elbeltagie, S.F.R. Khadri, Pandurang Choudhari (2021b), Estimation of crop and forest biomass resources in a semi-arid region using satellite data and GIS, Journal of the Saudi Society of Agricultural Sciences, 20(5), pp. 302-311.

2. Pande, C.B., Moharir, K.N., Khadri, S.F.R. et al. (2018).  Study of land use classification in an arid region using multispectral satellite images. Appl Water Sci 8, 123, https://doi.org/10.1007/s13201-018-0764-0.

3. Pande, C.B., Moharir, K.N. & Khadri, S.F.R. (2021b). Assessment of land-use and land-cover changes in Pangari watershed area (MS), India, based on the remote sensing and GIS techniques. Appl Water Sci 11, 96. https://doi.org/10.1007/s13201-021-01425-1 

Best of luck.

Author Response

Thank you for pointing to the references – we have added them to the paper.

Reviewer 3 Report

The study presented here raises the interesting issue of how to determine forest areas. 

My introduction lacks information on methods used in areas other than the USA. This would have given an overview of the different approaches not only in one country but also globally. 

Table 1 and 2, and Figure 7 What the authors meant when they wrote : "... but as something else in 2011 CDL ..."

The authors should perform a test of significance of differences where possible. 

Author Response

  1. My introduction lacks information on methods used in areas other than the USA. This would have given an overview of the different approaches not only in one country but also globally. 

Thank you for the suggestion, which is also echoed by Reviewer 2. We agree that our study has implications for forest assessments world-wide. We added such discussion in the revised Conclusions section.

  1. Table 1 and 2, and Figure 7 What the authors meant when they wrote : "... but as something else in 2011 CDL ..."

Thank you. This has been clarified in text.

  1. The authors should perform a test of significance of differences where possible. 

Since we define the differences between forestland representation in CDL and NLCD at the pixel level, conducting such tests within our approach is not possible. Thank you for the suggestion.

This manuscript is a resubmission of an earlier submission. The following is a list of the peer review reports and author responses from that submission.

Round 1

Reviewer 1 Report

Overall, the paper is well organized, written, and presented. It is also concise and properly cited. One could argue it is overly simplistic or too small in scope in this day and age, but that shouldn't take away from the paper as submitted.

Comments

  • You looked only at the NLCD and the CDL. I'm not sure when you work started but readers could ask about more contemporary products. Specifically, the LCMAP, LCMS, and GFW. Were these products considered?
    • https://www.usgs.gov/special-topics/lcmap
    • https://www.fs.usda.gov/rmrs/tools/landscape-change-monitoring-system-lcms
    • https://www.globalforestwatch.org/
  • Provide some background on the inherent uncertainties/errors that exist within the NLCD and CDL. Then describe if the differences you are seeing might just be because of the noise.
  • The arrows on the maps are confusing. Maybe just get rid of them.
  • Differences are documented but the reader is left with no real sense of what it all means. Close the paper with firm recommendations on how to use these products for there own use in tracking forest extent and change. In other words, is one product preferred? Should both be used? What are each's strengths and weaknesses of each?

Reviewer 2 Report

The authors explored the consistency of the two raster-based forest datasets. The topic is interesting. However, the current manuscript is more like a survey report rather than an academic paper. Neither new technology nor advanced knowledge was presented in this paper.

-Line 19. Statement of ‘2011-2020’ is not clear, because NLCD is only released in 1992, 2001, 2011 and 2016.

-The innovation of this study, if any, should be highlighted.

-Introduction must be improved. In spite of details of NLCD and CDL, current works based on these two datasets should be added, which was hardly referred in the manuscript.

-I suggest the authors to read and cite the following high-quality articles for improving their paper.

  1. Prestele R, Alexander P, Rounsevell MDA, et al. Hotspots of uncertainty in land-use and land-cover change projections: a global-scale model comparison. Global Change Biology. 2016;22(12):3967-83. doi:10.1111/gcb.13337
  2. Wulder MA, White JC, Niemann KO, Nelson T. Comparison of airborne and satellite high spatial resolution data for the identification of individual trees with local maxima filtering. International Journal of Remote Sensing. 2004;25(11):2225-32. doi:10.1080/01431160310001659252
  3. Ringvall A, Petersson H, Stahl G, Lamas T. Surveyor consistency in presence/absence sampling for monitoring vegetation in a boreal forest. Forest Ecology and Management. 2005;212(1-3):109-17. doi:10.1016/j.foreco.2005.03.002

-Line 103: https://nassgeo-103 data.gmu.edu/CropScape/, accessed on day-month-year. The same below.

- The authors should briefly introduce the study area, and explain why the area was selected.

-The consistency between NLCD and CDL was investigated. What is the actual accuracy of the two products?

-Figure 3 should be a map rather than a simple picture.

-Were there similar studies? Some references should be added to support the results/conclusion of this work in Discussion.

Reviewer 3 Report

  1. authors should be checked grammars and language errors also which Arc GIS version software used for this study, we should mentioned in the article.
  2.  Keywords is not confusing please revise as per the papers.
  3.  Authors should revised the introduction section because so many errors and uncomplete sentence found in this section. very poor language in the materials and methods.
  4. Figure 2 quality poor, authors should be improved.
  5. Results sections is a very short describe, authors should discuss more details about the results.
  6. conclusion section also very short please authors should be improved the overall this paper.
  7. I have checked whole paper but I have suggested for major revision of paper. 

Round 2

Reviewer 2 Report

More details of the data were added. However, I do not find any substantive amendment in the revised manuscript, especially methods and results. Please highlight any scientific or technical questions that answered in this study. By the way, I think Figure 5 is a table rather than a figure.